# Enhanced clinical assessment of hematologic malignancies through routine paired tumor and normal sequencing

Ryan N. Ptashkin[1,6,10], Mark D. Ewalt [1,10] ✉, Gowtham Jayakumaran[1,7], Iwona Kiecka[1], Anita S. Bowman [1], JinJuan Yao[1], Jacklyn Casanova [1], Yun-Te David Lin[1], Kseniya Petrova-Drus [1], Abhinita S. Mohanty[1], Ruben Bacares[1], Jamal Benhamida [1], Satshil Rana[1], Anna Razumova[1], Chad Vanderbilt [1], Anoop Balakrishnan Rema [1], Ivelise Rijo[1], Julie Son-Garcia[1], Ino de Bruijn [2], Menglei Zhu[1], Sean Lachhander[1], Wei Wang[1], Mohammad S. Haque[1], Venkatraman E. Seshan [3], Jiajing Wang[1], Ying Liu[1], Khedoudja Nafa[1], Laetitia Borsu[1], Yanming Zhang[1], Umut Aypar[1], Sarah P. Suehnholz[2], Debyani Chakravarty [1], Jae H. Park [4], Omar Abdel-Wahab [2,4], Anthony R. Mato[4], Wenbin Xiao [1], Mikhail Roshal[1], Mariko Yabe[1], Connie Lee Batlevi [4], Sergio Giralt[4], Gilles Salles [4], Raajit Rampal [1], Martin Tallman[4,8], Eytan M. Stein[4], Anas Younes [4,9], Ross L. Levine [2,4], Miguel-Angel Perales [4,5], Marcel R. M. van den Brink [2,4], Ahmet Dogan [1], Marc Ladanyi[1], Michael F. Berger [1,2], A. Rose Brannon [1], Ryma Benayed[1,9], Ahmet Zehir [1,9,10] ✉ & Maria E. Arcila [1,10] ✉

Genomic profiling of hematologic malignancies has augmented our understanding of variants that contribute to disease pathogenesis and supported development of prognostic models that inform disease management in the clinic. Tumor only sequencing assays are limited in their ability to identify definitive somatic variants, which can lead to ambiguity in clinical reporting and patient management. Here, we describe the MSK-IMPACT Heme cohort, a comprehensive data set of somatic alterations from paired tumor and normal DNA using a hybridization capture-based next generation sequencing platform. We highlight patterns of mutations, copy number alterations, and mutation signatures in a broad set of myeloid and lymphoid neoplasms. We also demonstrate the power of appropriate matching to make definitive somatic calls, including in patients who have undergone allogeneic stem cell transplant. We expect that this resource will further spur research into the pathobiology and clinical utility of clinical sequencing for patients with hematologic neoplasms.

Hematologic malignancies are characterized by the presence of complex and dynamic genomic changes that are now increasingly utilized to classify and define independent disease subsets. With the rapid adoption of next-generation sequencing technology, a multitude of recurrent somatic alterations in genes regulating cell growth, DNA repair, and differentiation have been identified; these contribute not only to the onset and progression of disease, but also to the development of relapse and resistance to therapy. Genetic profiling has

hence emerged as a key element in the workup of patients with hematologic malignancies, guiding patient management at various levels. While mutations in certain genes, such as *BRAF, CALR, JAK2* and *MPL*, have diagnostic utility in myeloid neoplasms, for example, others such as *CEBPA, DNMT3A, FLT3, IDH1, IDH2, KIT, NPM1*, and *TP53* have prognostic and/or therapeutic implications, particularly when determining whether a patient should undergo an allogeneic stem cell transplant[1,2]. Clinically, as the evidence and the repertoire of molecularly targeted therapies for hematologic malignancies continue to expand[3–5], so do the challenges and opportunities for molecular profiling to inform tumor classification, prognosis, disease monitoring, and treatment decisions.

Given the growing number of clinically relevant genetic alterations, it has become necessary to develop high throughput approaches for the genomic characterization of neoplasms in clinical practice. Unlike the workflows that have successfully provided prospective tumor molecular profiling of solid cancers at large scale[6–8], there are unique challenges to the evaluation of somatic alterations in hematologic malignancies. One distinct challenge is the lack of easily implementable sources of patient matched normal controls as comparators to confidently identify variants as distinctly somatic. The presence of leukemic contamination in buccal swabs and saliva, poor yield of DNA from hair follicles or nails and the extensive work required to sort normal cells or grow fibroblast cultures are all well-known challenges in clinical practice. Alternatively, unmatched interpretation brings its own challenges related to the discrimination of somatic and germline variants, especially given the large proportion of altered genes that do not have mutational hotspots or are not yet well-described. This precludes the reliance on publicly available databases for accurate curation of variants. Secondly, co-existing alterations that influence variant allele frequencies (VAF) [i.e., copy number alterations, copy neutral loss of heterozygosity (CN-LOH)] commonly occur, such that this metric cannot confidently guide the determination of somatic vs germline origin. These challenges are especially compounded in patients with a history of allogeneic hematopoietic stem cell (HSC) transplant, where determination of somatic vs germline and the source (host or donor) is often not possible.

Here, we show our experience addressing these unique challenges through the development and clinical experience of MSK-IMPACT Heme (Integrated Mutation Profiling of Actionable Cancer Targets for Hematologic malignancies), a comprehensive molecular profiling platform, utilizing hybridization capture and high coverage next-generation sequencing of paired tumor and normal tissues.

## Results

### Prospective clinical sequencing and utilization of different germline comparators

We developed MSK-IMPACT Heme to target 400 genes which are known to be involved in the pathobiology of hematologic neoplasms, are used for diagnosis and prognostication in hematological cancers, and are targets of experimental or approved therapeutic agents (Supplementary Table 1). We have previously described the application of paired tumor normal sequencing for patients with solid tumor malignancies to identify definitive somatic mutations of tumor origin[6,9–11]. To confidently identify somatic mutations in hematologic tumor cells, we used either saliva or nail clippings[12] as a source of germline DNA, since genomic material from whole blood may contain high levels of contaminating tumor cells and would not be suitable as a comparator (Fig. 1a, see Methods for details). During the analytical validation, mutation detection demonstrated 100% sensitivity and 100% specificity for 278 known mutations in 113 samples across a range of allele frequencies (range: 0.02–0.97) (Supplementary Fig. 1). Following approval from New York State Department of Health (NYS-DOH), between December 2016 and August 2019, we sequenced 2383 tumor samples, from 1937 patients, representing 85 different

hematological malignancies (Fig. 1b). Of these 2383 tumor samples, 1602 (67%) were sequenced with matched nail DNA, 664 (28%) with matched saliva, and 27 (1%) with both. For the 67 (3%) samples, from 48 patients, that were sequenced following allogeneic stem cell transplantation, both host and donor DNA derived from non-neoplastic were sequenced as a comparator (Fig. 1d).

We observed somatic tumor mutations in both saliva and nail at different levels based on disease modality (Fig. 1e). While nail DNA was most often purely germline, contaminating tumor DNA was observed with a VAF > 2% in 117 of 1295 (9%) patients and was enriched in chronic myeloid neoplasms, such as a myeloproliferative neoplasms (MPN, PMF, ET, and PV, 43 out of 170 patients, 25.3%), MDS (25 out of 132 patients, 19%), CMML (5 of 21 patients, 24%), and AML (16 out of 170 patients, 9%) (Fig. 1f). Of the 16 AML cases, the majority had evidence of an antecedent chronic myeloid neoplasms (*n* = 5) or history to suggest an evolving myeloid neoplasia, including prior chemotherapy/radiation exposure (*n* = 3), and/or persistent cytopenia (*n* = 2). Despite the presence of contamination, the variants detected in nail samples were found with high tumor:nail VAF ratios in virtually all cases (median 8; range 1.5–38), supporting the utility of nail control samples towards deciphering the germline versus somatic nature of variants detected in neoplastic patient samples. Somatic variants were rarely identified with VAFs >=10% in nail samples and were primarily confined to disease-defining alterations associated with loss of heterozygosity (LOH) in the tumor sample, such as *JAK2* and *TET2* in myeloproliferative neoplasms. These alterations were still easily identified as somatic variants owing to the retention of high tumor:nail VAF ratios (Fig. 1e). We detected 59 variants with a VAF > 2% in saliva controls from 31 patients, with the vast majority diagnosed with lymphoid neoplasms (90%) of T cell origin. The most frequently identified mutations were in *DNMT3A, TET2*, and *TP53*, which are commonly associated with clonal hematopoiesis and suggest the presence of a concurrent clonal myeloid process. While only a negligible number of patients with myeloid malignancies (*n* = 6/1,026) were sequenced with a saliva normal comparator, these saliva controls contained high levels of contaminating tumor DNA, up to 38% VAF (Fig. 1e–g). This finding is consistent with other studies[13], which suggest a limited role for saliva as a germline control in myeloid neoplasms. (Fig. 1e, g).

### Definitive identification of somatic variants

To highlight the importance of sequencing a matched germline comparator, we analyzed variant calls made in all targeted exonic regions of the MSK-IMPACT Heme panel resulting from 'unmatched' variant calling of these tumor samples against a pooled control sample composed of ten diploid blood samples (Supplementary Fig. 2, Supplementary Table 2). This analysis resulted in 48,248 variants that were properly filtered by the matched tumor-normal analysis pipeline, but otherwise passed criteria for clinical reporting, namely minimum VAF (0.05), variant sequence reads (10) and their absence from a panel of 25 curated normal samples, known to be lacking any hematologic malignancy. Of these, 27,611 (57%) were present in the gnomAD database with any population frequency >0.01, the primary recommendation for population database filtering from the joint consensus of AMP, ASCO, and CAP[14], and therefore annotated as putative germline variants that could be dropped in a tumor only analysis. Of the remaining 20,637 putative germline variants, 9,157 (44%) were present in COSMIC v94 database, and identified in 2,271 tumor samples, or 95% of our sequenced cohort with an average of 4 additional variants per sample. These represent germline variants that would have been incorrectly reported as somatic in an unmatched analysis, with potential adverse clinical implications. For instance, while specific mutations are not required in the FDA approval for hypomethylating agents (HMAs) in myeloid neoplasms, their presence has been associated with response to HMA treatment, and inaccurate reporting could alter choice of therapy[15]. In this analysis we identified a total of 54

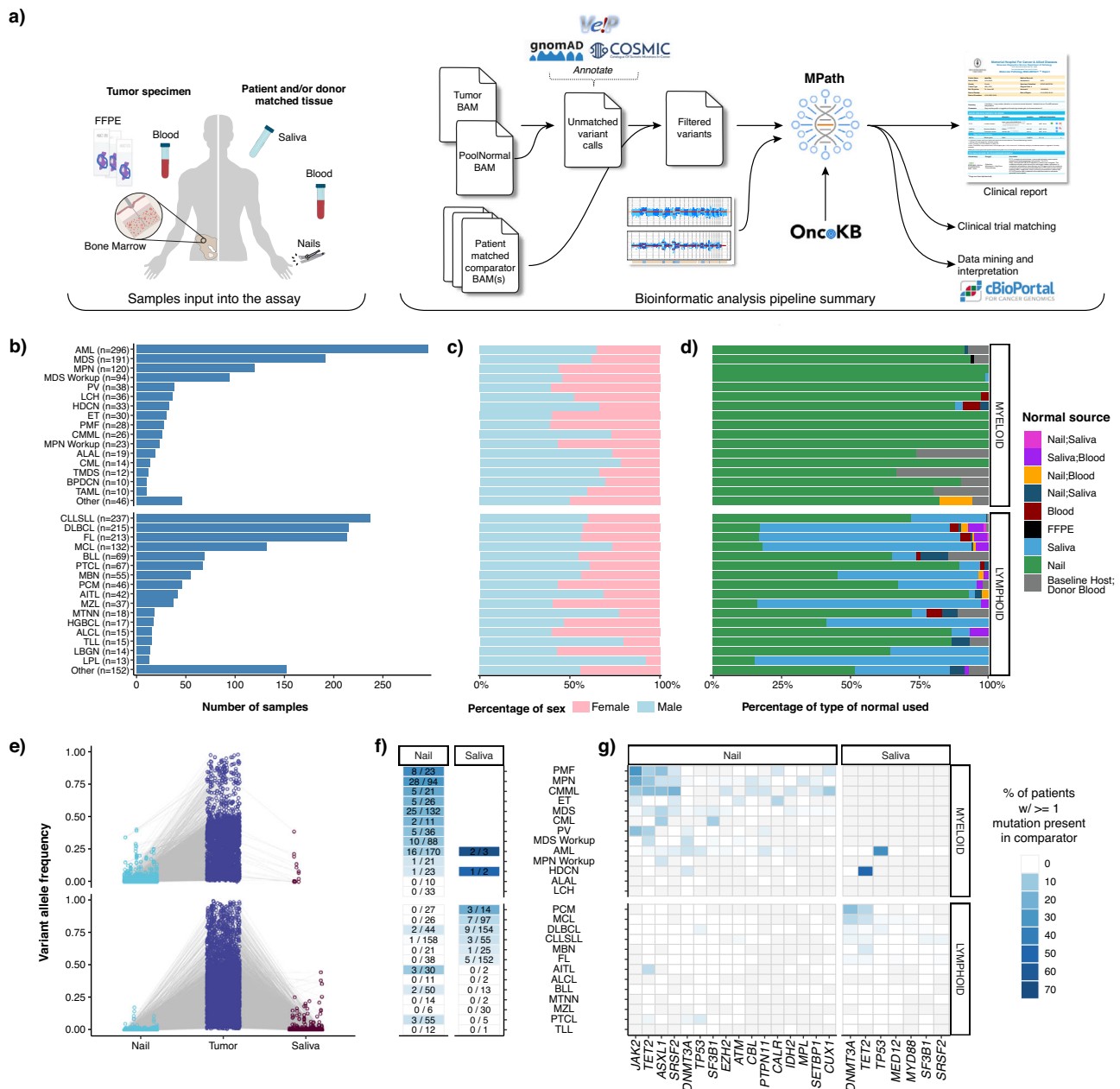

**Fig. 1 | MSK-IMPACT Heme workflow, sample distribution, and somatic mutation distribution in normal tissues. a** Overview of the MSK-IMPACT Heme clinical workflow. Distribution of (**b**) tumor types profiled by MSK-IMPACT Heme including (**c**) patient sex by tumor type, and (**d**) patient-matched normal sample type used for analysis. (**e**) Variant allele frequency (VAF) of somatic mutations in tumor and matched normal tissues. **f** Heatmap showing the percentage of patients with tumor somatic mutations observed in matched nail or saliva tissues. The first number in each cell indicates the number of patients where a tumor mutation is observed in the normal comparator, and the second number indicates the total number of patients profiled with the corresponding normal. **g** Heatmap showing the distribution of genes, where a somatic mutation is found in the tumor and the rate of observing the same variant in the matched normal, indicated with the color-scale. Gray cells indicate that for a given tumor type, either a somatic alteration was not detected in the tumor in that gene or the matched normal sample type (nail or saliva, shown above the heatmap) was not sequenced. Source data are provided as a Source Data file.

germline variants from myeloid neoplasms in genes associated with response to HMA treatment that could have been misattributed to being of somatic origin without a matched normal sample: *TET2* (n = 47), *DNMT3A* (n = 4), and *TP53* (n = 3) (Supplementary Table 3). In addition, in cases with equivocal morphology the presence of a variant is interpreted as evidence of a clonal process and may be used in support a malignant diagnosis such that inclusion of these germline variants could lead to erroneous diagnosis, overtreatment or, under the updated WHO and ICC classifications, could alter the diagnostic category (Supplementary Table 2). Lastly, the persistence of somatic alterations is often used to monitor response to therapy, therefore

misattribution of these alterations as somatic could lead to inaccurate monitoring results in follow-up samples.

Further, to identify prognostically and diagnostically important small- and large-scale somatic copy number alterations (SCNAs), we developed an algorithm (FACETS2n), which leverages coverage data from patient unmatched normal samples and combines with patient matched allele frequencies to estimate integer level copy number values as well as allelic imbalances such as copy-neutral loss of heterozygosity (CN-LOH) (see Methods). A comparison of results from high-density single nucleotide polymorphism (SNP) array and MSK-IMPACT Heme FACETS2n analysis from 64 clinical samples showed

highly concordant results with 92.9% sensitivity and 100% specificity. Discordant calls between SNP array and FACETS2n were attributed to sub-clonal calls made by SNP array and/or tumor fractions below the sensitivity of FACETS2n (less than 20%) as determined by serial dilution of a well characterized tumor sample. (Supplementary Tables 4–6).

Having established saliva and nail tissues as suitable controls for identifying true somatic SNVs and indels, we sought to further leverage paired tumor-normal matched sequencing data to assess the allele specific copy number alterations of this cohort. The identification of somatic copy number alterations (SCNAs), including gains and losses in chromosomal arms, has both diagnostic and prognostic implications for hematologic malignancies[16–18]. Historically, karyotype, fluorescent in-situ hybridization (FISH), and single nucleotide polymorphism (SNP) arrays have been used to detect clinically relevant SCNAs. The application of the FACETS2n algorithm to these sequencing data allowed the identification of focal amplifications and deletions as well as broad chromosomal arm level gains and losses. We detected focal copy number alterations in 854 patients (44.1%) whereas 1146 patients (59.2%) had a chromosome arm level copy number alteration detected. The identification of SNVs, indels and SCNAs in a single assay afforded efficiencies in tissue management and the ability to provide clinically actionable results from a single assay in a clinical setting.

### Use of host and donor normal controls to identify somatic alterations in the transplant setting

Confident identification of somatic variants in samples from relapsed patients in the post-transplant setting is a distinct challenge. By sequencing donor-derived DNA, we were able to confidently identify and remove donor germline polymorphisms in 47 out of 48 patients profiled following transplant. For one patient, a *TP53* variant identified post-transplant was also detected in the donor blood sample, but it was not possible to distinguish the germline vs somatic nature based on VAF alone in the tumor sample. In a second patient we identified a putative donor-derived somatic variant, *DNMT3A* p.R882C, likely of clonal hematopoiesis origin. To further demonstrate the utility of a unified analysis using both host and donor normal tissues, we present the case of a 37-year-old female who underwent allogeneic stem cell transplantation from an HLA-matched unrelated donor for the treatment of acute myeloid leukemia. A bone marrow biopsy was performed on day 98 post-transplant for assessment of suspected relapse, which was confirmed with 56% myeloblasts. Engraftment assessment by short tandem repeat analysis (STR) showed a chimeric status with 56% host component (Supplementary Fig. 3A). We performed MSK-IMPACT Heme on this relapse bone marrow using a pooled control sample as a comparator and called variants. To distinguish somatic mutations from germline polymorphisms, all variant calls were genotyped in the host nail and donor blood samples. Somatic mutations were defined as those with a variant allele fraction (VAF) of at least 0.02 in the bone marrow and not detected in host and donor samples. This approach allowed us to accurately distinguish all host and donor polymorphisms from somatic mutations in this chimeric patient where the range in VAF of host and donor-derived polymorphisms overlapped that of the true somatic mutations (range = 0.13–0.22) (Fig. 2a).

In addition to removing background polymorphisms, the use of FACETS2n enables more sophisticated local copy number analysis in the post-transplant setting. While computational methods have been developed to infer CN-LOH from SNP array data, both with and without an appropriate matched normal, these methods are impeded by false positives when using unmatched normals[19,20] and have not been optimized to analyze, or are not applicable to, samples from patients following allogeneic stem cell transplant chimeric patients due to the potential presence of heterozygous SNPs from more than one individual and unchanged integer copy number. To deal with these challenges, we adapted the FACETS[21] algorithm to use the intersection of heterozygous SNPs between baseline host and donor(s) samples to calculate variant allele log odds ratios with the post-transplant sample and determine regions of allelic imbalance genome wide (Supplementary Fig. 4). To illustrate the power of this approach, we present the case of a patient with a history of AML with a *FLT3* internal tandem duplication (ITD) mutation who underwent allogeneic stem cell transplant. *FLT3* ITD mutations, such as the 60 bp *FLT3*-ITD detected in this bone marrow (See Methods), are recurrent somatic alterations in AML and typically detected using PCR and capillary electrophoresis assays. (Supplementary Fig. 3B). Using DNA derived from patient nails and donor blood as baseline sample comparators to the post-transplant bone marrow biopsy, we were able to detect CN-LOH of chromosome 13q (Fig. 2b), indicating loss of the wildtype (WT) *FLT3* allele. This case illustrates the power of the joint utilization of matched patient and donor normal tissues to differentiate between somatic alterations and both host and donor-derived common polymorphisms, as well as to identify allele-specific copy number changes in patients after transplant.

### Profiling of sorted aberrant cell populations to increase diagnostic accuracy

The presence of multiple atypical or neoplastic populations in a sample is not uncommon in patients with hematologic malignancies. These may form part of a clonally heterogenous, single neoplastic process or may represent multiple synchronous neoplastic clones. Clinically, this difference is often difficult to tease out and patients may remain under- or mis-diagnosed and mismanaged. The use of flow sorting or other enrichment practices is a highly valuable approach and may be successfully performed to enrich very small populations for downstream analysis with our hybridization capture assay. To demonstrate the utility of analyzing flow sorted samples with MSK-IMPACT Heme, we highlight the case of a 72-year-old male undergoing diagnostic workup for angioimmunoblastic T cell lymphoma (AITL). Morphologic and immunophenotypic assessment of a bone marrow sample demonstrated low-level involvement by AITL (<5% by CD3/PD-1 immunohistochemistry) and the concurrent presence of a clonal plasma cell population, which accounted for 15% of cells on the aspirate smear and 1.9% of WBC by flow cytometry. Although clonal plasmacytosis has been reported in AITL[22,23], it remained unclear whether this represented a secondary neoplasm or a reactive expansion. Abnormal T cells and plasma cells were therefore sorted by flow cytometry (Supplementary Fig. 5) and submitted for mutational analysis at the direction of the hematopathologist reviewing the case to compare the mutation profiles among these compartments. Independent molecular profiling confirmed the two populations had distinct mutational profiles with the T cell population harboring *IDH2*, *RHOA*, *DNMT3A*, and *TET2* mutations[24,25], typical of AITL, while the plasma cells harbored *BRCA2*, *BTG*, *EPHA5*, *KMT2D*, and *SETD5* mutations (Fig. 3a). In addition, the two samples harbored unique copy number alteration profiles supporting the diagnosis of 2 separate neoplasms (Fig. 3b, c). Of note, only the *DNMT3A* and *TET2* mutations were identified in the unsorted marrow, suggesting that other mutations in the subpopulations were masked as an overall dilution effect in the bulk sample. While *DNMT3A* and *TET2* mutations have been reported to reside in both AITL and clonally related CH[26], which may account for the detection of these alterations in both the enriched T-cell and unsorted samples[27,28], the ability to sort and enrich samples is a powerful tool to interrogate mixed hematopoietic samples to assess clonal relatedness and understand the underlying biology of each population.

### Somatic genomic landscape

We identified 12,893 somatic mutations, 4231 gene level and 7566 broad chromosome arm level somatic copy number alterations from

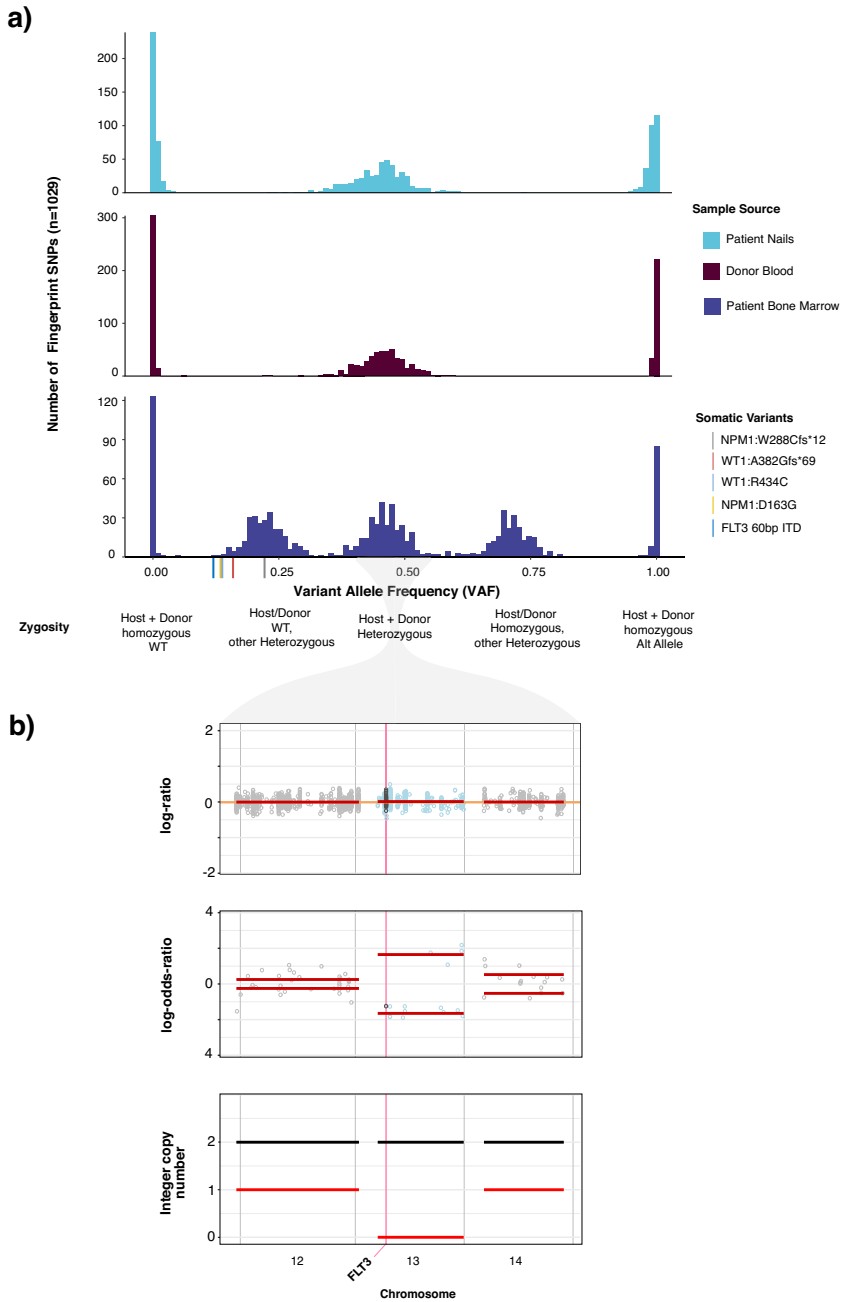

**Fig. 2 | Example patient case highlighting ability of IMPACT-HEME and donor/ host matching to detect complex mutations and allele-specific copy number in a post-transplant chimeric patient. a** The distribution of VAF of somatic mutations, host SNPs, and donor SNPs. **b** Detection of CN-LOH of chromosome 13q, including FLT3. Source data are provided as a Source Data file.

2290 samples. Implementation of the MSK-IMPACT Heme workflow enabled the characterization of complex tumor specimens, including flow-sorted samples and tumor samples from chimeric post-transplant patients. Somatic genomic alterations including nonsynonymous SNVs, indels, focal and chromosome arm level copy number alterations were identified in 1885 of 1937 patients (97.3%). A total of 1804 patients (93.1%) had at least one SNV or Indel identified (median = 4, range 0–191). The most commonly identified SNVs were in *KMT2D* ($n = 291$, 15%), *TP53* ($n = 288$, 15%), *TET2* ($n = 254$, 13%) and *CREBBP* ($n = 216$, 11%). (Fig. 4a) We observed broad, tumor purity corrected chromosome level alterations more commonly in lymphoid malignancies (69%, $n = 932/1357$) compared to myeloid neoplasms (37%, $n = 377/1026$; $p < 0.001$, Fisher's exact test). The most prevalent arm-level SCNAs in lymphoid neoplasms were +7p ($n = 157$, 12%), +18q

($n = 153$, 11%), +12q ($n = 148$, 11%), del 17p ($n = 215$, 16%), del 6q ($n = 196$, 14%), and del 13q ($n = 169$, 13%). For myeloid neoplasms, trisomy 8 ($n = 63$, 6%), +21q ($n = 27$, 3%) and +1q ($n = 18$, 2%) were the most prevalent broad gains, while del 7q ($n = 59$, 6%), del 17p ($n = 42$, 4%), del 5q ($n = 40$, 4%), and del9p ($n = 40$, 4%) were the most common broad chromosomal losses (Fig. 4c). These findings have been well described in myeloid neoplasia and, in particular, del5q and del7q are considered sufficient to render a diagnosis of MDS, even in the absence of morphologic dysplasia[17]. We further compared biological pathways based on the genes in these deleted regions. Lymphoid neoplasms were significantly enriched for deletions in genes of the following pathways: p53 (18% vs 5%, $q$ (FDR-corrected $p$-value) $=1.15 \times 10^{-15}$, two-proportion Z-test), immune modulation (11% vs 0%, $q = 9.25 \times 10^{-18}$), NOTCH signaling (10% vs 2%, $q = 1.47 \times 10^{-10}$), chromatin modifiers (9% vs 5%,

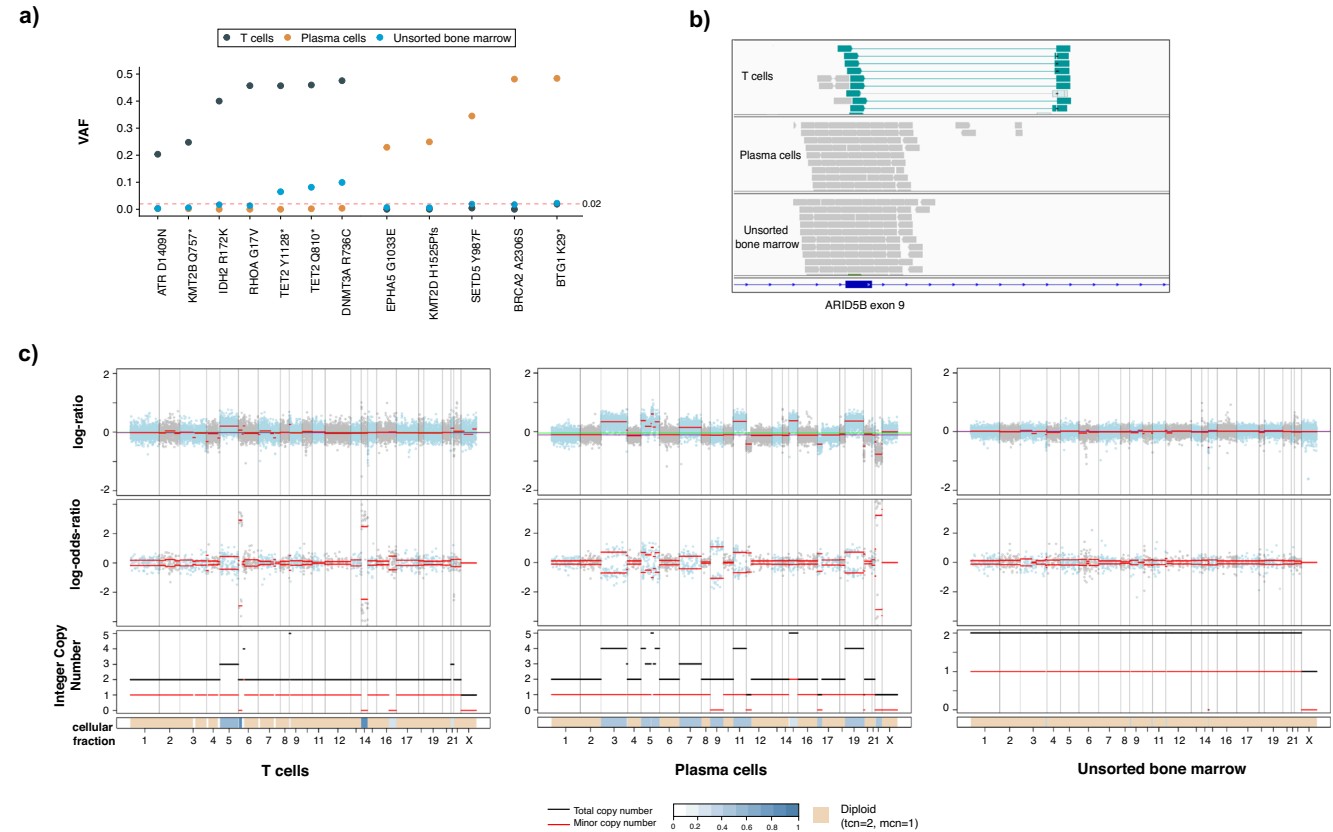

**Fig. 3 | Example case of utility for flow-sorted genomic analysis in a patient with AITL and an atypical plasma cell population. a** Somatic mutational profiles of T cells, plasma cells and unsorted bone marrow highlighting distinct mutation patterns in different populations. **b** ARID5B structural rearrangement (inversion) detected only in the T cell population. **c** The unique somatic copy number alteration profiles of sorted T cells and plasma cells compared to unsorted bone marrow further support that these populations are clonally distinct. Source data are provided as a Source Data file.

$q = 4.18 \times 10^{-2}$), DNA damage response (7% vs 0%, $q = 9.51 \times 10^{-12}$), and NF-kB signaling (6% vs 1%, $q = 2.26 \times 10^{-7}$). The most prevalent focal copy number alterations in myeloid neoplasms were deletions of *TP53* ($n = 42$, 5%), *JAK2* ($n = 44$, 5%), *FLT3* ($n = 17$, 2%), *TET2* ($n = 16$, 2%), and EZH*2* ($n = 16$, 2%).

In addition to gene level copy number alterations, FACETS2n enables accurate assessment of allele-specific copy number state, including copy neutral loss of heterozygosity (CN-LOH). CN-LOH was identified in 433 samples (19%) and, similar to global copy number changes, was more frequently noted in lymphoid neoplasms ($n = 294/1357$, 22%) compared to myeloid ($n = 139/1026$, 14%, $p < 0.001$, Fisher's exact test) including FL ($n = 103$, 48%), DLBCL ($n = 92$, 43%), and HGBCL ($n = 6$, 35%). In myeloid malignancies, CN-LOH was observed in acute leukemias including AML ($n = 36$) and BLL ($n = 6$), or chronic myeloid neoplasms including PMF ($n = 14$, 50%), CMML ($n = 9$, 35%), and PV ($n = 13$, 34%). The most frequent chromosome arm level CN-LOH events were identified in 6p ($n = 81$), 9p ($n = 65$), 16p ($n = 50$), 9q ($n = 48$), 16q ($n = 46$), 17q ($n = 46$), 15q ($n = 44$), 19p ($n = 44$), 13q ($n = 43$), and 17p ($n = 41$). Interestingly, CN-LOH has been shown to be a mechanism of HLA Class 1 loss in cancer and may underlie the 6p aberrations noted here[29].

Through the integration of SNV/Indel variants and SCNAs, several genes were identified to harbor one mutated allele in conjunction with LOH of the wild type allele. This phenomenon has been well-documented to occur with *TP53* across tumor types, *ATM* in lymphoid neoplasms, *JAK2* in myeloproliferative neoplasms, and *TET2* in myeloid neoplasms[30–37]. We found similar results with these genes (*TP53* $n = 153$, *ATM* $n = 53$, *JAK2* $n = 33$, and *TET2* $n = 30$) as well as several other genes. In particular, within FL and DLBCL, the following

genes were frequently affected by two hits via mutation and LOH: *TNFRSF14* ($n = 76$), *CREBBP* ($n = 71$), *TNFAIP3* ($n = 30$), and *B2M* ($n = 29$). We also identified genes which harbored multiple somatic variants in a single neoplastic sample, which may reflect bi-allelic inactivation, multiple subclones, or aberrant somatic hypermutation. In myeloid malignancies, multiple alterations were noted in *TET2* ($n = 142$, including 24/43 or 56% of AITL samples) and *DNMT3A* ($n = 35$), while in mantle cell lymphoma, *ATM* ($n = 14$, 10%) frequently harbored multiple mutations. FL and DLBCL showed multiple mutations in the same patient of *KMTD* ($n = 101$), *CREBBP* ($n = 39$), and *HIST1H1E* ($n = 20$), in addition to aberrant somatic hypermutation of *BCL2* ($n = 58$), *PIM1* ($n = 41$), and *SOCS1* ($n = 22$). *TP53* harbored multiple mutations across lymphoid and myeloid malignancies ($n = 82$) (Fig. 4b).

## Mutational signatures

The application of DNA sequencing in conjunction with advances in mathematical models have aided the discovery and understanding of the mutational processes that underlie the acquired somatic variants of cancer genomes[38–40]. In clinical tumor profiling, the deciphering of mutational signatures can aid diagnosis, disease prognosis, and treatment decisions[41–44]. However, identification of mutational signatures has occurred mostly in solid tumor cohorts, mainly due to the lower levels of somatic mutation in blood cancers relative to solid tumors[38,45]. In the MSK-IMPACT Heme cohort, we calculated the tumor mutation burden (TMB, see Methods) for all samples (range 0–192.9, median 3.7 mut/Mb) (Fig. 5a). Relative to myeloid malignancies, lymphoid tumors were characterized by a higher TMB (mean 8.7 vs 3.0 mut/Mb, $p < 0.001$). For the 261 tumors (11%) with elevated tumor

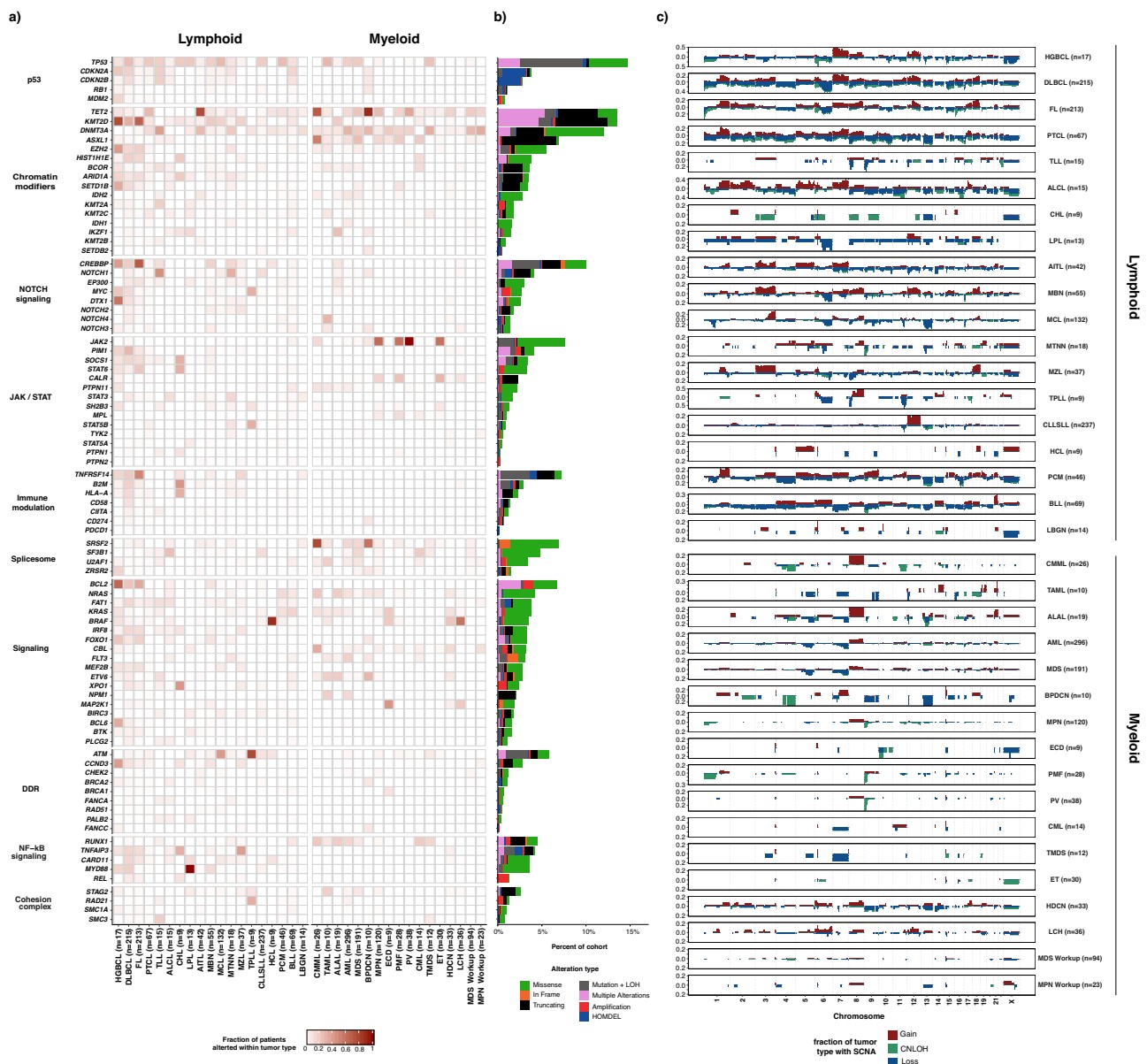

**Fig. 4 | Somatic genomic landscape of hematologic malignancies profiled by MSK-IMPACT HEME. a** Recurrent somatic alterations across common tumor types and pathways in lymphoid and myeloid neoplasms. **b** Bars indicate the percentage of cases harboring different classes of genomic alterations, with integration of mutation and allele specific copy number status. **c** Genome-wide somatic copy number (SCNA) profiles in main tumor types. Source data are provided as a Source Data file.

mutation burden (>12.9 Mut/Mb, see Methods), all synonymous and nonsynonymous single nucleotide variants were decomposed into COSMIC v3.1 SBS signatures with the inclusion of recently described MMRd signatures[40,46] (See Methods).

We identified tumors with mutational processes attributable to activation-induced cytidine deaminase (AID) activity, DNA polymerase eta, mismatch repair (MMR) deficiency, exposure to ultraviolet light (UV), chemotherapy treatment, apolipoprotein B editing complex (APOBEC), and clock-like mutational processes (Fig. 5b). Mature B-cell neoplasms with elevated tumor mutation burden (n = 231) displayed dominant mutational signatures associated with genome instability as mediated by AID and the error-prone DNA polymerase eta in conjunction with clock-like mutational processes[47–49]. We observed ultraviolet light exposure as a dominant signature in cutaneous T-cell lymphomas (n = 11) in addition to DLBCL tumors from two patients, for which clinical histories indicated that these two tumors likely originated near the skin.

Nine tumors from seven patients exhibited a dominant MMR signature, including all four relapse BLL tumor samples with elevated tumor mutation burden. Of the nine tumors with MMR signatures, only one DLBCL sample did not have a clear alteration in the MMR pathway. This case had a lower TMB (20 Mut/Mb) and estimated tumor purity (28%) relative to other MMR tumors and only 43% of mutations attributed to MMR signature. Other signatures attributed to the mutational profile of the tumor were associated with AID, polymerase eta, and clock-like mutational processes. We also observed somatic *MLH1* alterations in the two BLL samples with highest mutation burden concurrent with *MSH6* frameshift variants and heterozygous loss of *MSH2/6*. In the first sample, we detected CN-LOH of *MLH1*, and in the other, a splice variant (c.790+1G>A) previously reported to result in exon 9–10 skipping and reported as a pathogenic germline variant in many individuals with a family history of Lynch-syndrome associated tumors exhibiting microsatellite instability[50,51]. No somatic *PMS2* alterations were detected in tumors with dominant MMR signature.

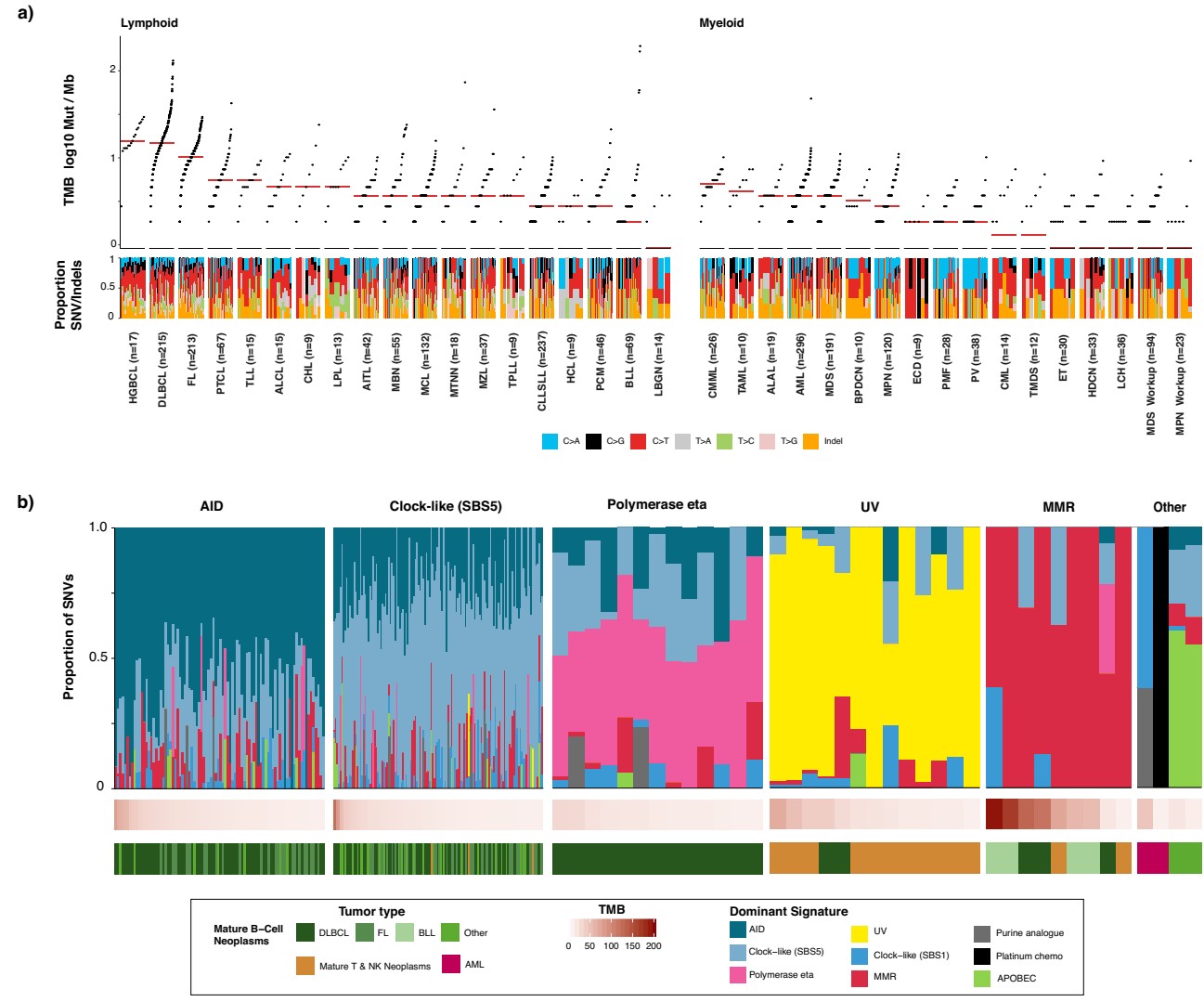

**Fig. 5 | Mutation signature analysis in MSK-IMPACT Heme cohort. a** Prevalence of somatic mutations across main tumor types and the mutational spectra. The median TMB for each tumor type is indicated by a red line. **b** Mutational signatures, sorted by dominant signature for the 261 tumors with elevated mutation burden (>12.9 Mut/Mb). Source data are provided as a Source Data file.

Both AML samples with elevated mutation burden had dominant mutational signatures for chemotherapy that corresponded with their treatment histories. The two samples with elevated mutation burden and a dominant APOBEC signature were plasma cell myelomas, which has been previously shown to be a poor prognostic indicator[52]. Taken together, we show the ability of mutation signature analysis from targeted sequencing of hematologic cancers with elevated TMB to identify underlying mutational processes, with potential to impact patient management using these data.

## Clinical actionability

We also sought to assess the clinical utility of prospective molecular profiling to guide patient management using OncoKB (http://oncokb.org), an expert curated precision oncology knowledge base. OncoKB annotates the oncogenic effect and clinical implications of somatic molecular alterations and has recently expanded to include alterations in hematologic malignancies[53,54]. Key to OncoKB is its level of evidence system that annotates molecular variants based on the level of evidence that the alteration is either a predictive biomarker of drug sensitivity or important in informing diagnosis or prognostication. By classifying patient samples by the highest level of evidence assigned to detected variants in that sample, we found that 10.6% of patients

profiled had at least one potential clinically actionable alteration, defined as carrying ≥1 alterations assigned an OncoKB level of evidence 1-3B[55]. and 71.5% had an oncogenic alteration (Fig. 6a). In tumor only analysis using a threshold of >0.01 MAF in gnomAD, numerous false-positive oncogenic calls would have been inappropriately included: an additional 14 variants in 14 cases (0.6% of cohort) would have been called as oncogenic and 551 variants in 485 cases (19% of cohort) would have been called as likely oncogenic. These false positives were removed using appropriately matched normals. Moreover, 43% of patients had at least one alteration with a diagnostic (Dx) or prognostic (Px) significance as defined by the OncoKB Dx and Px levels of evidence[54] or SCNAs detected by IMPACT-Heme meeting IPSS-R criteria or prognostic indicators in CLL (del13q, trisomy 12, del11q, and del17p). Of note, this analysis reflects an underestimate of actionability at a disease level in this cohort, as the MSK-IMPACT Heme assay does not include targets for the detection of actionable gene fusions or rearrangements which may be addressed in future panel design iterations. Instead, transcript fusion detection is accomplished by a companion RNA-based NGS assay[56].

In MDS, the International Prognostic Scoring System–Revised (IPSS-R) is the current standard for patient risk stratification, which relies on clinical parameters of cytopenias, bone marrow blast

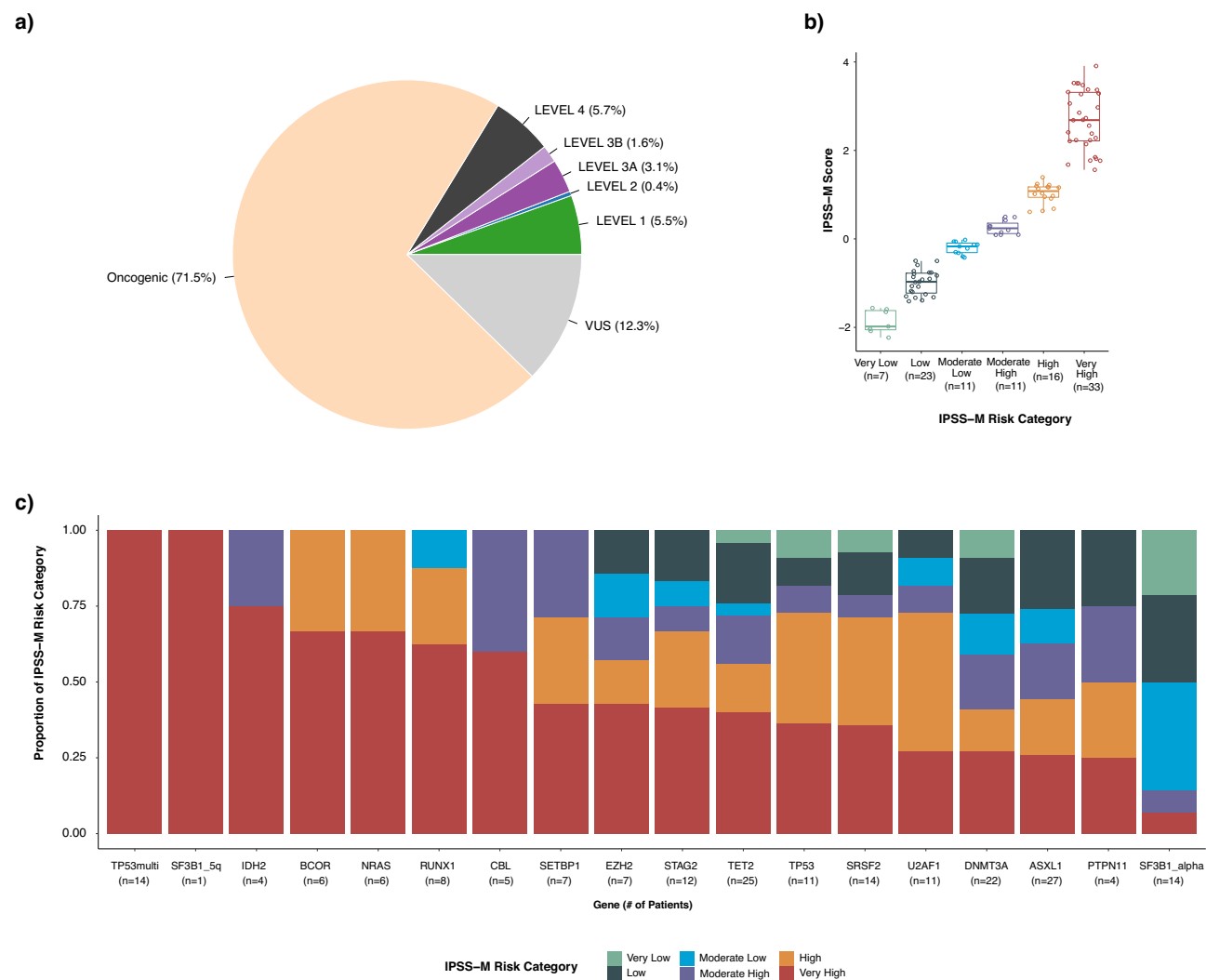

**Fig. 6 | Clinical actionability of MSK-IMPACT Heme results. a** Percentage of samples across all tumor types that harbor a mutation considered clinically actionable according to the OncoKB therapeutic levels of evidence. **b** Distribution of IPSS-M algorithm calculated scores and risk categories identified in the 101 cases in the MSK-IMPACT Heme cohort. In the box plots, the central line represents the median; the box corresponds to 25–75% quartiles; the upper whisker extends to the largest value no farther than 1.5× IQR; and the lower whisker extends from the 25% quartile to the smallest value no farther than 1.5× IQR. **c** The number of patients in the MSK-IMPACT Heme MDS cohort with the given genomic alteration and their stratification into IPSS-M risk categories. Source data are provided as a Source Data file.

percentage, and cytogenetic features, but does not consider gene mutations[16]. The recently described IPSS-Molecular (IPSS-M) model includes these features in combination with genomic profiling to improve risk stratification (https://mds-risk-model.com)[57]. Here, we applied the IPSS-M model to the 101 patients with MDS for whom we also had the required clinical and cytogenetic data to stratify each patient into IPSS-M risk categories. After application of this algorithm, 32.3% (n = 33) of cases were classified with a risk category of Very High, 15.8% (n = 16) as High, 10.9% (n = 11) as Low, and 6.9% (n = 7) as Very Low. (Fig. 6b). Evaluation of variants using a tumor-only approach and a threshold of >0.01 MAF in gnomAD would have altered (incorrectly inflated) the IPSS-M score of 8 patients (8% of MDS cohort) as follows: 'High' -> 'Very High' (n = 1); 'Moderate High' -> 'High' (n = 2); 'Moderate Low' -> 'Moderate High' (n = 2); 'Low' -> 'Moderate Low' (n = 2); 'Very Low' -> 'Low' (n = 1). Combining somatic copy number alterations detected by MSK-IMPACT Heme with conventional karyotyping (G-banding) and FISH allowed for a more sensitive detection of copy number alterations, mainly due to the detection of CN-LOH via IMPACT-Heme. Amongst those patients categorized as very high risk, we identified 14 patients with multiple hits to *TP53*, two patients with

multiple *TP53* variants and 12 patients with a single variant and loss of heterozygosity. For patients with a single *TP53* hit, 36% (n = 4) were classified as very high risk. We also demonstrate the ability of IMPACT-Heme to distinguish between subgroups of *SF3B1* altered patients, with a single patient identified with mutated *SF3B1* and isolated del5q versus 14 patients identified with *SF3B1* alpha (lacking co-mutations in *BCOR, BCORL1, NRAS, RUNX1, SRSF2,* or *STAG2*) and associated with favorable outcomes[57] (Fig. 6c).

## Discussion

We, herein, report the experience of a large institution-wide, prospective clinical sequencing effort to guide the diagnosis, prognosis, therapy selection and future monitoring of patients across the spectrum of hematologic malignancies. As we previously demonstrated in solid tumors[6], we now show that this type of enterprise-scale sequencing of neoplastic and matched normal samples is feasible in hematologic cancers, including highly complex cases of patients following hematopoietic stem cell transplantation and those with multiple concurrent malignancies. For a comprehensive matched assessment, turnaround times may be variable and, in this cohort,

generally ranged from 2-3 weeks. Shorter turnaround times are possible depending on the timeframe to obtain the tumor-normal pairs, further highlighting the importance of a multidisciplinary approach for workflow optimization. While this is a clinically actionable timeframe in a large proportion of cases, the implementation of alternate assays for rapid assessment of key genetic alterations may be needed as a bridging step in select cases. Through this comprehensive effort, we demonstrate the utility of our approach, going beyond the narrow assessment of selected genes in a tumor only sequencing model, which may be incomplete and ineffective for patient management. As the costs of sequencing and data analysis continue to fall, we believe that a similar matched approach will be applied to whole genome sequencing (WGS) for all cancer patients. In this future model, inclusion of germline controls will be even more critical than in a targeted panel where the normal variation of the targets is better understood. We have generated an extensive collection of manually reviewed mutations and SCNAs in 2,384 samples from 1,937 patients in 85 detailed tumor types. This cross-malignancy dataset will support explorations of driver alterations across all blood cancers to support discovery of rare and unanticipated clinically actionable alterations. With continuing growth in the realm of precision therapeutics, this data set will prove a transformative resource for identifying novel biomarkers to inform prognosis and predict response and resistance to therapy. This includes the further definition of putative passengers which may, in fact, have important functional and cooperative roles in driving cancer. In an unmatched model, a very high proportion of private SNP's are classified as passengers or VUS's, necessarily so because they are not yet part of well characterized variants in any of the current publicly available databases. In contrast to solid tumors where the primary focus of genomic profiling has been the selection of targeted therapy for key single genetic drivers, the aims of genomic profiling in hematologic cancers are heavily invested in refining a diagnosis and providing prognostic information, with therapy selection often supported by the former. Broad genomic profiling provides a more accurate diagnosis and risk stratification of individual patients at the time of diagnosis and may also predict response and/or outcomes after selected treatments. For instance, *TP53* mutations are consistently associated with shorter survival after allogeneic stem cell transplantation and somatic mutations in epigenetic pathways (*TET2*, *IDH1/2*, *WT1*, and *DNMT3A*) may confer increased sensitivity to hypomethylating agents[15,58,59]. Somatic mutations may require reassessment to update individual risk after treatment, at the time of significant clinical changes or before disease-modifying treatments. Our approach to testing, incorporating routine sequencing of appropriate control samples, enables the unequivocal identification of somatic genetic variants in a way that is scalable even in the context of an allogeneic transplant. It also allows the determination of donor-derived variants which may necessitate monitoring in both the recipient and donor for subsequent development of disease. Additionally, a separate analysis of the normal controls would also facilitate the assessment of key germline events that are relevant to hematologic malignancies. Although this was not included here due to a lack of patient consent for dedicated germline analysis and reporting, this is the topic of a manuscript in preparation with a more recent data set.

At the same time, as our understanding of the biology of hematologic malignancies has continued to expand, compounds targeting proteins or signaling pathways disrupted by recurrently mutated genes have become available, notably inhibitors to *EZH2* in follicular lymphoma and *FLT3*, *IDH1*, and *IDH2* in AML[60,61]. One emerging area of study in hematologic cancer is the study of mutation signatures. In solid tumors, mutation signatures such as MMR deficiency and TMB correlate with response to immune checkpoint inhibitors[62,63]. In contrast to solid tumors, hematologic malignancies tend to have lower levels of somatic mutation[6,38], which may account for their relatively disappointing response to immune checkpoint inhibition[64–68]. The unambiguous identification of somatic alterations via the use of a matched normal affords a more accurate assessment of TMB in these neoplasms. Our identification of a small subset of patients with high TMB and/or MMR signatures suggests that these patients could be biological outliers and should be considered for trials of checkpoint inhibition based on these signatures.

In addition to the biological insights and potential for therapeutic targeting afforded by our approach, there is also an opportunity for improved patient monitoring. There has been increasing interest in designing assays for monitoring minimal/measurable residual disease (MRD) following treatment across hematologic malignancies[69,70]. In spite of this interest; however, it is unclear if suitable markers are available for all patients and some guidelines only specify molecular targets for select patients[71]. To address this shortcoming, some groups propose approaches which include the use of any somatic alteration as a potential target for monitoring[72,73]. These approaches highlight the power of a fully matched sample at initial tumor genotyping and the pitfalls of inaccurate somatic/germline assignment. By removing germline variants from reporting through genotyping of candidate variants in matched normal tissue(s), we are able to better identify appropriate markers for MRD assessment and prevent false positive calls. We have demonstrated through our tumor only analysis that these confounders occur in up to 95% of samples and could thus significantly limit the power of an "any variant" MRD approach.

While this study represents a foray into the power of broad-scale genomic analysis in hematologic malignancy, additional work remains for the field of clinical genomic analysis to reach its full potential to improve patient care. Other groups have begun to explore even broader testing for a subset of hematologic malignancies[74], and it is our hope that work such as that presented here and by other groups will serve as a precedent for increased genomic profiling in blood cancer. The best approach to rapidly achieve these goals is through sharing of these datasets across institutions and establishing broad collaborations. To this end, we have deposited our full data set into the cBioPortal for Cancer Genomics (https://www.cbioportal.org/study?id=heme_msk_impact_2022). With continued testing and data sharing, it is our belief that broad genomic assessment will support understanding the pathobiology of, identifying novel drug targets for, and improving non-invasive monitoring for response in all patients with blood cancer.

## Methods

### Patient consent and accrual

This study complies with all relevant ethical regulations. Informed consent for the molecular profiling of patient tumors was obtained under protocol NCT01775072 "Tumor Genomic Profiling in Patients Evaluated for Targeted Cancer Therapy." The protocol was approved by the Institutional Review Board at Memorial Sloan Kettering Cancer Center and written consent was obtained from all patients. Following consent, either archival or new tumor samples were obtained. The selection of appropriate matched normal was determined after a review of the patient clinical history and tumor diagnosis. OncoTree (http://www.cbioportal.org/oncotree/), an institutional tumor classification system was used to ensure consistent specimen annotation. Matched saliva was prioritized for lymphoid neoplasms owing to the ease of specimen collection and processing and the known paucity of lymphoid components in the samples. Patient-matched nail tissue is requested for all myeloid neoplasms due to the high level of neoplastic myeloid cells in the patient whole blood and saliva. Patients that had previously undergone hematopoietic stem cell transplantation were sequenced with pre-transplant host and/or donor normal specimens, dependent on engraftment status and tissue availability.

### Assay design and validation

We designed custom DNA probes targeting 1.08 Mb of the human genome corresponding to all protein-coding exons and the adjacent

20 bp of intronic sequence of 400 key oncogenes and tumor suppressor genes implicated in hematologic malignancies, including all genes that are targetable by approved and experimental therapies being investigated in clinical trials at our institution.

To determine the accuracy, precision, and sensitivity of the assay, we analyzed DNA from 113 unique tumor DNA samples confirmed to be positive for SNVs and Indels in 50 exons of 20 cancer genes previously genotyped or sequenced in our clinical laboratory[11]. These samples comprised 11 tumor types from blood, bone marrow, and FFPE tissues (Supplementary Fig. 1A) and had been previously genotyped or sequenced in our clinical laboratory and were confirmed to be positive for mutations by multiple methods. The objective of the accuracy study was to assess the ability of the assay to detect mutations previously confirmed by the reference method in the tested sample. All 278 variants, from 52 exons of 20 genes, were successfully detected with the IMPACT-Heme assay (Supplementary Fig. 3B). In addition, there was high reproducibility amongst replicates from both inter- and intra-assay experiments. Samples positive for SNVs and indels were tested in triplicate in the same sequencing run and on different days in two additional sequencing pools (Supplementary Fig. 3C). To determine the analytical sensitivity of the assay, we performed serial dilutions of tumor samples with known variants and determined the VAF at each dilution as output from the variant calling pipeline. The detection limit for low-frequency variants was approximately 2% (Supplementary Fig. 3D). The ability to detect somatic copy number alterations was demonstrated with samples previously characterized by clinically validated SNP array platforms. MSK-IMPACT Heme was validated and approved for clinical use by the New York State Department of Health Clinical Laboratory Evaluation Program. Following approval, MSK-IMPACT Heme testing was implemented in the clinic to identify genomic alterations that could potentially inform diagnosis and treatment decisions.

### MSK-IMPACT Heme sequencing and analysis workflow

MSK-IMPACT Heme is a custom hybridization capture-based assay for the detection of single nucleotide variants (SNVs), small insertions and deletions (Indels), and somatic copy number alterations. Genomic DNA extraction was performed on the Chemagic STAR instrument (Hamilton) from peripheral blood, bone marrow, saliva, or formalin-fixed, paraffin-embedded (FFPE) tumors and patient-matched normal samples using the Chemagen magnetic bead technology (PerkinElmer). FFPE tissues were deparaffinized using mineral oil followed by digestion with the proteinase K enzyme. Extraction of genomic DNA from nail samples was performed by utilizing both physical and chemical digestion techniques. 10−25 mg of nail clippings were pulverized using high-speed agitation and centrifugation with zirconium beads in a BeadBlaster instrument (Benchmark Scientific, NJ) followed by chemical digestion with an adapted protocol using QIAamp® DNA investigator kit (Qiagen).

DNA samples were normalized to yield 50−250 ng input and diluted with Tris-EDTA (diluted from 100X solution, Fisher Scientific Catalog Number BP1338-1) to a total volume of 55 µl on the Biomek FX Laboratory Automation Workstation (Beckman Coulter), before undergoing shearing on the Covaris instrument. Sequence libraries were prepared through a series of enzymatic steps, including shearing, end-repair, A-base addition, ligation of barcoded sequence adaptors, and low-cycle PCR amplification (Kapa Biosystems). To enable multiplexed captures, tumors and matched normal samples were combined into pools of approximately 28 libraries, utilizing custom-designed biotinylated probes (Nimblegen). The captured DNA fragments were subsequently sequenced on an Illumina HiSeq2500 as paired-end 100-base pair reads.

An automated data management system monitored the sequencers, initiating the analysis pipeline upon completion of the sequencing run. Sequence reads were aligned to the human genome (hg19) using BWA MEM (version 0.7.5a). ABRA (version 0.92) was employed to realign reads around indels to reduce alignment artifacts, and the Genome Analysis Toolkit (version 3.3-0) was used to recalibrate base quality scores. Duplicate reads were identified and marked for removal, resulting in BAM files that were utilized for variant candidate discovery.

We implemented a custom analysis pipeline (see below) to integrate the analysis of any number of normal samples with a given tumor and provide a reliable assessment of somatic alterations, even in post-transplant chimeric patients. Copy number alterations were assessed using FACETS2n, an allele-specific copy number analysis pipeline for next-generation sequencing data, adapted from the FACETS algorithm[21] to allow the incorporation of multiple normal samples for normalization and determination of allelic imbalance in tumor samples, even those from chimeric patients. All genomic variants called by the analysis pipeline were loaded into MPath, an in-house genomic variant database and user interface that facilitates the manual review of variants and their assessment for therapeutic, diagnostic, and prognostic implications with OncoKB[53]. Through the incorporation of variant allele fraction (VAF) in tumor and normal tissues, patient clinical history, and annotated population frequencies[9], we were able to eliminate variants with low sequencing quality and those of patient and/or donor germline origin.

### SNV/Indel calling

Variant calling was performed in paired sample mode using BAM files generated for the tumor sample and the pooled normal control sample processed with each sequencing run. Indel realignment of sequencing reads was performed with ABRA2 (version 2.13)[75] prior to variant calling to resolve soft-flipped bases to insertions and deletions commonly missed by standard analysis workflows, such as FLT3-ITDs. To the union of calls made by MuTect (version 1.1.4)[76], VarDict (version 1.4.6)[77], and Somatic Indel Detector (version 2.3)[78], the genotypes from the patient-matched normal sample(s) were incorporated and subjected to automated filtering to generate a complete list of somatic mutation calls, including SNVs and short and long indels. By incorporating the genotype information for patient and donor DNA of non-neoplastic origin, we were able to eliminate variant calls attributed to the germline present in tumor specimens. In detail, all variant calls require a VAF in the tumor ≥ 5 times that of an unmatched normal, a minimum of 20 total reads, 5 alt reads, at least 1% VAF, and presence in less than 20% of our standard normal samples. Furthermore, hotspot sites required an alt allele depth of 8 reads and a VAF ≥ 2%. Non-hotspot sites require more stringent secondary filtering of at least 10 alt reads, VAF ≥ 5%, and VAF in the matched normal and tumor ≤ 35%. Each alteration identified by the pipeline was annotated with The Ensembl Variant Effect Predictor (VEP)[79] to be compliant with Human Genome Variation Society (HGVS, http://varnomen.hgvs.org) standards and then manually reviewed to ensure that no false positives were reported.

### Copy number analysis

Genome-wide total and allele-specific copy number states were calculated for all tumor samples using the open-source R package FACETS2n (v0.3.0). Library-specific coverage biases that stem from differences between tumor (FFPE, blood, and bone marrow) and normal (Nails, Saliva, Blood) tissues may result in log ratios with high levels of noise when calculated with matched normal samples. With FACETS2n, a single unmatched normal is selected from a pool of high-quality normal samples previously processed and sequenced with the MSK IMPACT-Heme assay. These normal samples were selected to have a representation of males and females from a variety of tissue types and with different insert size distributions. A single tumor

sample was compared to multiple control normal samples to derive distinct log-ratio sets. The normal sample with the lowest sum-squared log-ratio served as the best reference diploid genome comparator for the tumor sample[11]. The logOR of the variant-allele count in tumor versus patient-matched normal, an unbiased estimate of allelic copy ratio, was calculated for all heterozygous SNPs (alt allele freq between 0.25 and 0.75) in the patient-matched normal. For patients sequenced following allogeneic stem cell transplantation, logOR was limited to the subset of heterozygous SNPs common to the patient baseline normal sample and all donor samples. For the calculation of integer copy numbers, we utilized a two-pass implementation whereby a low-sensitivity run (cval = 150) first determines copy number log-ratio corresponding to diploidy. The copy number state of individual genes was determined by a run with higher sensitivity for focal events (cval= 75). The following gene level SCNAs were retained for analysis: amplifications (integer copy number ≥ 5 without whole genome doubling or > 6 with whole genome doubling), homozygous deletions, and heterozygous losses that co-occurred with an SNV or Indel. Broad chromosome arm level copy number gains (integer copy number ≥ 3 without whole genome doubling or > 5 with whole genome doubling) and losses were retained for analysis if they comprised at least 50% of the chromosome arm.

We evaluated the accuracy, sensitivity and reproducibility of MSK-IMPACT Heme in detecting somatic copy number alterations in a validation study of 11 select and clinically relevant regions: chromosomes 3q, 5q, 7q, 8, 11q, 12, 13q, 17p,19, 20q, and the single gene locus TP53. A total of 64 clinical samples were evaluated by both IMPACT-Heme and snp-array, using the results of snp-array as the set of true positive copy number alterations. In these clinical samples, somatic copy number alterations were detected with 92.9% sensitivity and 100% specificity. Three samples with copy number alterations from the validation set were studied over three different sequencing runs for inter-assay reproducibility in addition to being studied in triplicate in the same run for the intra-assay(precision). Concordant results were obtained for all cases in both intra- and inter-assay reproducibility studies. The sensitivity of FACETS2n was evaluated using one FFPE sample from a DLBCL patient with known 12p amplification and 13q loss. Five serial dilutions using DNA from patient-matched FFPE normal tissue were prepared (Original, 50%, 25%, 12.5% and 6.25%).

### TMB calculation

Tumor mutation burden was calculated as the number of nonsynonymous and synonymous SNVs and Indels per megabase of genome targeted by the MSK-IMPACT Heme panel (1.0837 Mb). We used the distribution of TMB across all tumors to identify highly mutated cases with the formula: median cohort TMB + 2*IQR. Those tumors with a TMB > 12.9 Mut/Mb were classified as TMB High and analyzed for mutational signatures.

### Mutational signatures

Mutational signatures were assessed for the 261 tumor samples with elevated mutation rates using all synonymous and nonsynonymous SNVs and SigProfiler software[38,40]. In order to limit inter-signature bleeding that stems from difficult to decipher flat signatures[40,80] and elucidate the mutational processes that contribute to individual cancer genomes, we first performed de novo extraction of single base substitution (SBS) signatures on both individual tumor types and groups of tumors originating from either lymphoid or myeloid lineages. Discovered signatures were decomposed into COSMIC v3.1 SBS signatures with the inclusion of recently described MMRd signatures[46]. We then estimated the contribution of each signature to individual cancer genomes using a nonlinear convex optimization programming solver, SigProfilerAssignment. The set of known signatures applied to each genome was determined from those identified from de novo extraction in the same tumor type or cellular lineage[40].

### IPSS-M

Clinical parameters of percentage bone marrow blasts, hemoglobin levels, and platelet counts were curated for 101 patients with an MDS diagnosis. We then compiled the somatic genomic alterations (SNVs, Indels, SCNAs) and pathologist-reviewed assessment of cytogenetic results to derive IPSS-R cytogenetic risk categories, identify complex karyotype, and encode the gene and chromosome level binary variables as input to the IPSS-M algorithm to derive the IPSS-M risk score[57].

### Detection of FLT3-ITDs

De novo detection of FLT3 internal tandem duplications (ITDs) using NGS data was performed by adding an indel realignment step to aligned BAM files using ABRA2[75] which incorporates high-quality soft-clipped reads into the generation of contigs that represent variation from the reference genome. ITDs that were resolved via indel realignment were then identified as part of the somatic variant calling pipeline with either the SomaticIndelDetctor and/or VarDict algorithms[77,78].

### Reporting summary

Further information on research design is available in the Nature Portfolio Reporting Summary linked to this article.

## Data availability

The raw sequencing data for the MSK-IMPACT analysis is protected and cannot be broadly available due to privacy laws; patient consent to deposit raw sequencing data was not obtained. All results derived from the analysis of clinical sequencing data (mutations, copy number alterations, and structural variants) are publicly available. Analysis of germline alterations was not performed due to lack of patient consent for dedicated germline analysis and reporting. The minimal clinical and somatic alteration data (including mutations and allele specific copy number calls) necessary to replicate the findings in the article are publicly available on cBioPortal: https://www.cbioportal.org/study?id=heme_msk_impact_2022 and have been deposited to https://github.com/mskcc/MSK_IMPACT_HEME. Source data are provided with this paper.

## Code availability

Analysis code for in-house developed pipeline modules is made available on Github. Facets2n: https://github.com/mskcc/facets2n[81].

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

## Acknowledgements

The authors gratefully acknowledge Nicole Degroat, Christine England, Sandy Naupari, Jake Bakas, Yingjuan Xu, Hina Patel, Srushti Kakadiya, Jaclyn Rutter, Justina Almodovar, Daviana Martinez-Osorio, Brandon Gray, Hun Jae Jung, Nelio Chaves, Jada Barbee, Kizzia Perez, Aqib Abass, and Shadia Islam for their important contributions. This research was funded in part through the NIH/NCI Cancer Center Support Grant P30 CA008748.

## Author contributions

M.F.B., R. Benayed, and M.E.A. designed the panel and the assay. M.F.B., A.R.B., R.Benayed, and A.Z. developed the assay. R.N.P., G.J., A.S.B., Y.-T.D.L., A.S.M., S.R., V.E.S., A.R.B., and A.Z. developed the bioinformatics pipeline. J.H.P., O.A.-W., A.R.M., C.L.B., S.G., G.S., R.R., M.T., E.M.S., A.Y., R.L.L., M.-A.P., and M.R.M.v.d.B. collected specimens for assay valida-tion. R.N.P., G.J., I.K., A.S.B., J.C., Y.-T.D.L., A.S.M., R. Bacares, S.R., A.R., A.B.R., I.R., J.S.-G., I.d.B., S.L., W.W., M.S.H., J.W., K.N., L.B., Y.Z., U.A., M.F.B., A.R.B., R.Benayed, A.Z., and M.E.A. generated and interpreted the validation data. R.N.P., G.J., I.K., A.S.B., J.Y., J.C., Y.-T.D.L., K.P.-D., A.S.M., R.Bacares, J.B., S.R., A.R., C.V., A.B.R., I.R., J.S.-G., I.d.B., M.Z., S.L., W.W., M.S.H., J.W., Y.L., K.N., L.B., Y.Z., U.A., S.P.S., D.C., W.X., M.R., M.Y., C.L.B., S.G., G.S., R.R., M.T., E.M.S., A.Y., R.L.L., M.-A.P., M.R.M.v.d.B., A.D., M.L., A.R.B., R.Benayed, A.Z., and M.E.A. generated and interpreted the clinical data. R.N.P., M.D.E., and A.Z. performed the analyses for the manuscript. R.N.P, M.D.E, A.Z., and M.E.A. wrote the manuscript with input from all authors.

## Competing interests

The authors declare the following competing interests: Ryan N. Ptashkin is an employee of C2i Genomics. Gowtham Jayakumaran is an employee of Guardant Health. Chad Vanderbilt reports intellectual property rights and equity interest in Paige.AI, Inc. Menglei Zhu has received advisory or consulting fees from Leica Biosystems. Jae Park has received advisory or consulting fees from Allogene, Amgen, Artiva, Autolus, BMS, Curocel,

Incyte, InnatePharma, Kite Pharma, Kura Oncology, Minerva, Novartis, Pfizer, PrecisionBio, Servier, and served on a data monitoring committee for Affyimmune, BrightPharma and Intellia. Omar Abdel-Wahab has served as a consultant for H3B Biomedicine, Foundation Medicine Inc, Merck, and Janssen; is on the Scientific Advisory Board of Pfizer Boulder, Envisagenics Inc., and AIChemy; and has received prior research funding from H3B Biomedicine, Loxo/Lilly, and Nurix Therapeutics. Anthony Mato has received research funding from TG Therapeutics, Pharmacyclics, AbbVie, Johnson and Johnson, Astra Zeneca, DTRM BioPharma, BeiGene, Genentech, Genmab Janssen, LOXO, Nurix, Octopharma, Pfizer and has received honoraria from TG Therapeutics, Pharmacyclics LLC, AbbVie, Adaptive Biotechnologies, Johnson and Johnson AcertaDTRM BioPharma, Nurix, AstraZeneca BeiGene Genentech Janssen LOXO, Curio, Dava, Octopharma, Genmab, BMS, Medscape, PER, PerView. Mikhail Roshal has received research funding from Roche, NGM, and Beat AML and has served on an advisory board and holds stock/stock options in Auron Therapeutics. Mariko Yabe has served as a consultant for Janssen R&D. Connie Batlevi has received advisory or consulting fees from ADC Therapeutics, AbbVie, Bristol-Myers Squibb, Dava Oncology, Defined Health, Epizyme, Gerson Lehrman Group, Juno Therapeutics, Karyopharm, Kite Pharmaceuticals, LifeSci Capital, LLC, Medscape, MorphoSys AG, NeuroAxis LLC, Seattle Genetics, Skipta LLC, TG Therapeutics, Inc, Touch Independent Medical Education Ltd, and Treeline Biosciences, Inc., and reports ownership/equity interest in Bristol-Myers Squibb, Moderna, Inc., Novavax, Pfizer, Inc., Regeneron Pharmaceuticals, Inc., and Viatris Inc. Sergio Giralt has received advisory or consulting fees from Amgen, CSL Behring, Caladrius, Celgene, Ceramedix, ExpertConnect, GlaxoSmithKline, Janssen Research & Development, LLC, Karyopharm, Kite Pharmaceuticals, Magnolia Innovation, Novartis, Omeros, Pfizer, Inc., Physicians' Education Resource, Sanofi US Services Inc., TRM Oncology, and Xcenda. Gilles Salles has received advisory or consulting fees from Abbvie, Beigene, Bayer, BMS/Celgene, Epizyme, Genentech/Roche, Genmab, Incyte, Ipsen, Janssen, Kite/Gilead, Loxo, Milteniy, Molecular Partners, Morphosys, Nordic Nanovector, Novartis, Rapt, Regeneron, and Takeda, and reports equity interest in Owkin. Raajit Rampal has received advisory or consulting fees from Constellation, Incyte, Celgene/BMS, Novartis, Promedior, CTI, Jazz Pharmaceuticals, Blueprint, Stemline, Galecto, PharmaEssentia, AbbVie, Sierra Oncology,Servier, and Disc Medicines; and research funding from Incyte, Constellation, and Stemline. Eytan Stein has received advisory or consulting fees from AbbVie, Agios Pharmaceuticals, Bristol-Myers Squibb, CTI BioPharma Corp., Calithera, Celgene, Epizyme, Genentech, Genesis Therapeutics, Gilead Pharmaceutical, Janssen Pharmaceuticals, Inc., Jazz Pharmaceuticals, Kronos Bio, Inc, Kura Oncology, Neoleukin Therapeutics, Inc., Novartis Pharmaceuticals Corporation, PinotBio, Inc., Servier, Syndax, and Takeda Millennium, and reports ownership/equity interests in Auron Therapeutics, Inc. Anas Younes is an employee of AstraZeneca. Ross Levine is on the supervisory board of Qiagen and is a scientific advisor to Imago, Mission Bio, Syndax. Zentalis, Ajax, Bakx, Auron, Prelude, C4 Therapeutics and Isoplexis for which he receives equity support; receives research support from Ajax and Abbvie; has consulted for Incyte, Janssen, Morphosys and Novartis; and has received honoraria from Astra Zeneca and Kura for invited lectures. Miguel Perales reports honoraria from Abbvie, Allovir, Astellas, Bristol-Myers Squibb, Caribou Biosciences, Celgene, Equilium, Exevir, Incyte, Karyopharm, Kite/Gilead, Merck, Miltenyi Biotec, MorphoSys, Novartis, Nektar Therapeutics, Omeros, OrcaBio, Takeda, and VectivBio AG, Vor Biopharma; serves on DSMBs for Cidara Therapeutics, Medigene, Sellas Life Sciences, and Servier, and the scientific advisory board of NexImmune; has ownership interests in NexImmune and Omeros; and has received institutional research support for clinical trials from Incyte, Kite/Gilead, Miltenyi Biotec, Nektar Therapeutics, and Novartis. Marcel van den Brink has received advisory or consulting fees from Ceramedix, DKMS, Da Volterra, Garuda Therapeutics, GlaxoSmithKline, LyGenesis,

Inc., Vor Biopharma, Pluto Immunotherapeutics, Rheos Medicines, Inc, Seres Therapeutics, Frazier Healthcare Partners, and Notch Therapeutics and has ownership/equity interests in Pluto Immunotherapeutics, Seres Therapeutics, ThymoFox, Inc., and Notch Therapeutics and is in a fiduciary role/position for DKMS, holds intellectual property rights with Juno Therapeutics and Seres Therapeutics, and has received royalties from Wolters Kluwer. Ahmet Dogan has received advisory or consulting fees from Incyte, Loxo Oncology, and Physicians' Education Resource and has received research funding from Roche and Takeda. Marc Ladanyi has received advisory or consulting fees from Takeda Oncology, Janssen Pharmaceuticals, AstraZeneca, ADC Therapeutics, Paige.AI, Merck, Bayer, and Lilly Oncology and has received research funding from Loxo Oncology, Helsinn Therapeutics, Merus NV, Elevation Oncology, and Rain Therapeutics. Michael Berger has received advisory or consulting fees from AstraZeneca, Eli Lilly and Company, and PetDx, Inc. A. Rose Brannon has ownership/equity interests in Johnson and Johnson. Ryma Benayed is an employee of AstraZeneca. Ahmet Zehir is an employee of AstraZeneca. Maria Arcila has received advisory or consulting fees from Axis Medical Education, Clinical Education Alliance, LLC, Merck Sharp & Dohme, PeerView Institute for Medical Education (PVI), Physicians' Education Resource, RMEI Medical Education, LLC, and Roche. The following Authors declare no competing interests: Mark D. Ewalt, Iwona Kiecka, Anita S. Bowman, JinJuan Yao, Jacklyn Casanova, Yun-Te David Lin, Kseniya Petrova-Drus, Abhinita S. Mohanty, Ruben Bacares, Jamal Benhamida, Satshil Rana, Anna Razumova, Anoop Balakrishnan Rema, Ivelise Rijo, Julie Son-Garcia, Ino de Bruijn, Sean Lachhander, Wei Wang, Mohammad S. Haque, Venkatraman E. Seshan, Jiajing Wang, Ying Liu, Khedoudja Nafa, Laetitia Borsu, Yanming Zhang, Umut Aypar, Sarah P. Suehnholz, Debyani Chakravarty, Wenbin Xiao, and Martin Tallman.

## Additional information

[1]Department of Pathology and Laboratory Medicine, Memorial Sloan Kettering Cancer Center, New York, NY, USA. [2]Human Oncology & Pathogenesis Program, Memorial Sloan Kettering Cancer Center, New York, NY, USA. [3]Department of Epidemiology and Biostatistics, Memorial Sloan Kettering Cancer Center, New York, NY, USA. [4]Department of Medicine, Memorial Sloan Kettering Cancer Center, New York, NY, USA. [5]Department of Medicine, Weill Cornell Medical College, New York, NY, USA. [6]Present address: C2i Genomics, New York, NY, USA. [7]Present address: Guardant Health, Palo Alto, CA, USA. [8]Present address: Lurie Comprehensive Cancer Center, Northwestern University, Chicago, IL, USA. [9]Present address: Oncology R&D, AstraZeneca, New York, NY, USA. [10]These authors contributed equally: Ryan N. Ptashkin, Mark D. Ewalt, Ahmet Zehir, Maria E. Arcila. ✉e-mail: ewaltm@mskcc.org; ahmet.zehir@astrazeneca.com; arcilam@mskcc.org

