## [Peer Review File · Nature Communications]

Editorial Note: This manuscript has been previously reviewed at another journal that is not operating a transparent peer review scheme. This document only contains reviewer comments and rebuttal letters for versions considered at Nature Communications. Mentions of the other journal have been redacted.

REVIEWER COMMENTS

Reviewer #1 (Remarks to the Author):

This is a resubmission of a manuscript previously reviewed at [Redacted].

In general, the authors have satisfactorily addressed most of the points made in the original review, with a few exceptions.

The authors should still show how often adding germline testing changes the ICC/WHO classification (asked for in the previous review) and not just IPSS-M scores.

The analysis of gnomAD MAF thresholds is still problematic. Using COSMIC as a gold standard for somatic mutations is fraught with error. Most of the COSMIC data comes from tumor only sequencing, and is full of rare SNPs that have been incorrectly designated as 'somatic' mutations. The AMP/ASCO/CAP guidelines for the 1% MAF cut-off is based on 2014 lab survey data, when molecular testing was still in its infancy—no one doing tumor only sequencing in 2023 uses a 1% MAF cut off. The new guidelines (under development) will suggest a much lower MAF cut-off. For that reason, showing the effects of lower MAF cutoffs (at least in the supplement) will be useful for readers going forward.

The authors argue that the lack of germline testing limits the ability to do MRD testing. This is not really correct. It's easy to identify germline variants in serial tumor only data, since the VAFs are stable at 50%.

Finally, cheap, fast, reliable whole genome sequencing is rapidly on the way for clinical applications; the cost of a 30x genome on the new Illumina Novoseq X platform will be about 325 dollars, and will soon fall even further. Several recent papers have reported on whole genome workflows in clinical settings, which will ultimately replace all platform tests (since no private reagents are needed, no capture reagents, universal approaches that are not test site-specific, etc.). Exome and panel sequencing were developed a decade ago when sequencing was expensive, and they were never intended to be the long

term gold standard. The authors don't even mention this approach, but they definitely should consider the movement towards WGS in the clinic in the discussion.

Reviewer #2 (Remarks to the Author): Expert in computational genomics

In this manuscript, Drs Ptashkin, Ewalt, Zehir and Arcila describe their diagnostic genomic experience from a large prospective cohort of haematological malignancies. The authors do a comprehensive job of highlighting the difficulties of performing tumour-only genomic analysis, and the impact of different sources of matched normal DNA. Several important points are made regarding the precious nature of the samples and the desire to perform comprehensive genetic analysis from a single assay. The introduction of false positive somatic variants from the tumour-only analysis is well made and should form a valuable contribution for others seeking to conduct precision medicine studies in haematological conditions. I would classify all of my recommendations as minor.

Minor suggestions

1. The data presented in Figure 2 was incomplete for highlighting how the donor and host SNPs are identified, partitioned, and use to create a clean patient-specific profile. Consider also presenting the noisy CNV profile fit without subtracting the donor, and strengthening the confidence that the donor (and not the patient) germline variants are accurately removed.
2. Line 278, what triggers the flow cytometry analysis in the workflow?
3. Line 347, compound heterozygosity is usually the combination of two [small] mutations on different alleles, not the combination of mutation and LOH. I recommend this be described as hemizyosity, or more colloquially, as a second hit.
4. Line 379, how many mutations minimum were required for this analysis. Provide (in silico) experimental justification for the choice of >12.9 Mut/Mb
5. Line 384: why did the authors not investigate HRD mutations, as I expect these would be prevalent in the cohort, and some studies suggest that there are several mutation signatures
6. Line 565, describe the characteristics of the panel. Total size, all exons, how much of the introns are included...
7. Line 575: was the accuracy study completed by researchers blinded to the expected results?
8. Line 585: what is the LOD of CNVs? Is this the same for deletions and amplifications?
9. Line 634: how are duplicate variants resolved in the variant union procedure? For example, variant VAF will likely differ between callers

10. Line 652: More methodological description of the FACETS2n algorithm is required, as this is a major component of the paper. The R package is well documented, and installs, but is lacking a LICENSE file, technically rendering it currently unusable. Please provide example data and supporting files so that the package can be rapidly tested

11. Line 737 The URL is dead, ensure this is active:
https://www.cbioportal.org/study/summary?id=heme_msk_impact_2022

12. Line 742 looks like a URL but is not.

Reviewer #3 (Remarks to the Author):

The authors respond to my concerns about the limited additional value of paired tumor-normal sequencing (vs tumor-only), by explaining that private SNPs or putative passengers may be of potential clinical relevance. Whilst this remains possible and the examples the authors cite are valid (tracking the tumor, mutational signatures etc), these are not demonstrated in their study.

So whilst the manuscript describes a robust and affective platform, the technical or clinical advances demonstrated here are limited.

NCOMMS-23-14700A

“Enhanced clinical assessment of hematologic malignancies through routine paired tumor:normal sequencing.”

Response to Reviewers

Reviewer #1 (Remarks to the Author):

This is a resubmission of a manuscript previously reviewed at [Redacted].

In general, the authors have satisfactorily addressed most of the points made in the original review, with a few exceptions.

The authors should still show how often adding germline testing changes the ICC/WHO classification (asked for in the previous review) and not just IPSS-M scores.

The analysis of gnomAD MAF thresholds is still problematic. Using COSMIC as a gold standard for somatic mutations is fraught with error. Most of the COSMIC data comes from tumor only sequencing, and is full of rare SNPs that have been incorrectly designated as ‘somatic’ mutations. The AMP/ASCO/CAP guidelines for the 1% MAF cut-off is based on 2014 lab survey data, when molecular testing was still in its infancy—no one doing tumor only sequencing in 2023 uses a 1% MAF cut off. The new guidelines (under development) will suggest a much lower MAF cut-off. For that reason, showing the effects of lower MAF cutoffs (at least in the supplement) will be useful for readers going forward.

Response: We agree with the reviewer that the 1% MAF cut off is probably too relaxed. We have included the table of lower MAF cut-offs and the number of mutations that pass these filters as a supplemental table 2 and also attached below for reference. We have also included additional discussion in the manuscript around this point. We hope this data will be useful for the new guideline preparation.

gnomAD MAF Threshold	# Non-somatic variants <gnomAD threshold	# Non-somatic variants <gnomAD threshold that would change	% Cases with Non-somatic variant that would result in change to ICC/WHO	# Non-somatic variants <gnomAD threshold in COSMIC	Average # COSMIC variants <gnomAD threshold per sample	% Cases with a COSMIC variant <gnomAD threshold
--	--	---	--	--	---

		ICC/WHO classification*	classification*	v94		
1%	20,637	42	1.8%	9,157	4.0	95.3%
0.5%	15,946	42	1.8%	5,958	2.8	89.5%
0.1%	10,523	42	1.8%	3,121	2.0	68.5%

The authors argue that the lack of germline testing limits the ability to do MRD testing. This is not really correct. It's easy to identify germline variants in serial tumor only data, since the VAFs are stable at 50%.

Response: While we agree that it is possible to infer germline variants based on the longitudinal assessment of VAF's across several samples, based on our own experience, this may lead to many errors given that it remains an assumption. VAF can be affected by many factors, including alterations in gene copy numbers and variations related to coverage and technology. More importantly, in transplant patients this is simply not possible due to the variable proportions of both host and donor components. In our practice, we very often see patients who were sequenced in outside institutions, who come in with a list of "mutations" our clinical teams expect us to track and that prove to be germline events that were not filtered. The use of a normal control not only facilitates the assessment in large panels but enables the unequivocal monitoring of tumor specific events that would not be possible otherwise, even in the context of transplant.

Finally, cheap, fast, reliable whole genome sequencing is rapidly on the way for clinical applications; the cost of a 30x genome on the new Illumina Novoseq X platform will be about 325 dollars, and will soon fall even further. Several recent papers have reported on whole genome workflows in clinical settings, which will ultimately replace all platform tests (since no private reagents are needed, no capture reagents, universal approaches that are not test site-specific, etc.). Exome and panel sequencing were developed a decade ago when sequencing was expensive, and they were never intended to be the long term gold standard. The authors don't even mention this approach, but they definitely should consider the movement towards WGS in the clinic in the discussion.

Response: We concur with the reviewer that whole-genome sequencing (WGS) technology will have a place in the clinical management of patients with hematological malignancies. However, there is still a long way to go before this becomes a reality outside of a few select academic centers or large clinical research organizations. This is due to the costs associated with sequencing platforms and the technical expertise required to analyze WGS data for decision-making purposes. Platforms like Oxford nanopore could close this gap even further, and large language models like GPTs could enable easier interpretation of results in the future. WGS sequencing will absolutely require a matched normal sample to confidently identify somatic alteration spectrum This has also been addressed in the discussion.

Reviewer #2 (Remarks to the Author): Expert in computational genomics

In this manuscript, Drs Ptashkin, Ewalt, Zehir and Arcila describe their diagnostic genomic experience from a large prospective cohort of haematological malignancies. The authors do a comprehensive job of highlighting the difficulties of performing tumour-only genomic analysis, and the impact of different sources of matched normal DNA. Several important points are made regarding the precious nature of the samples and the desire to perform comprehensive genetic analysis from a single assay. The introduction of false positive somatic variants from the tumour-only analysis is well made and should form a valuable contribution for others seeking to conduct precision medicine studies in haematological conditions. I would classify all of my recommendations as minor.

Minor suggestions

1. The data presented in Figure 2 was incomplete for highlighting how the donor and host SNPs are identified, partitioned, and use to create a clean patient-specific profile. Consider also presenting the noisy CNV profile fit without subtracting the donor, and strengthening the confidence that the donor (and not the patient) germline variants are accurately removed.

Response: We thank the reviewer for this suggestion and have added an additional figure to the manuscript as eFigure4. This figure highlights the genome-wide allele-specific copy number profile of a post-transplant patient obtained using (a) the baseline host germline reference and (b) the intersection of heterozygous SNPs between baseline host and baseline donor germline reference samples. Using a baseline reference from only one individual results in the inability to accurately identify regions of allelic imbalance due to host-donor chimerism in the post-transplant setting. By utilizing baseline germline reference samples from both the host and donor, we are able to confidently identify regions of allelic imbalance, including copy neutral loss of heterozygosity.

2. Line 278, what triggers the flow cytometry analysis in the workflow?

Response: All sequencing studies are reviewed in the context of existing ancillary studies and clinicopathologic information. Flow cytometry results are always reviewed as part of the workflow of analysis of patient samples. When small populations are identified which would be difficult to analyze through bulk sequencing, the hematopathologist reviewing the case will initiate flow sorting to enrich for the population(s) of interest prior to sequencing.

3. Line 347, compound heterozygosity is usually the combination of two [small] mutations on different alleles, not the combination of mutation and LOH. I recommend this be described as hemizygoty, or more colloquially, as a second hit.

Response: We thank the reviewer for this suggestion. The language has been updated in the text of the manuscript.

4. Line 379, how many mutations minimum were required for this analysis. Provide (in silico) experimental justification for the choice of >12.9 Mut/Mb

Response: We have previously shown the high correlation of TMB from targeted sequencing data to whole exome sequencing, informing that the panel sequencing results are representative of genome-wide processes:

<https://www.ncbi.nlm.nih.gov/pmc/articles/PMC5461196/>. The subset of cases with high TMB were identified as detailed in the Methods: “We used the distribution of TMB across all tumors to identify highly mutated cases with the formula: median cohort TMB + 2*IQR. Those tumors with a TMB >12.9 Mut/Mb were classified as TMB High and analyzed for mutational signatures.”

5. Line 384: why did the authors not investigate HRD mutations, as I expect these would be prevalent in the cohort, and some studies suggest that there are several mutation signatures

Response: As HRD was not identified in our de novo extraction and fitting procedure for any individual tumor type nor class of tumor (myeloid, lymphoid), it was not included for mutational signature analysis in this cohort, see Methods.

6. Line 565, describe the characteristics of the panel. Total size, all exons, how much of the introns are included...

Response: We have updated this section of the text to include additional information on panel size and regions that are targeted.

7. Line 575: was the accuracy study completed by researchers blinded to the expected results?

Response: Laboratory processing and bioinformatics analysis of validation samples was performed by individuals and analysis pipelines without prior knowledge of the expected results.

8. Line 585: what is the LOD of CNVs? Is this the same for deletions and amplifications?

Response: CNV LOD was determined to be 20% tumor fraction for both deletions and amplification, as determined by serial dilution of a tumor sample previously characterized by SNP array. This is highlighted in the text lines 188-194 and Supplemental Tables 4-6. Although it is possible for high level amplifications to be detected at lower tumor fractions, integer copy number estimations below 20% tumor fraction suffer a loss of accuracy.

9. Line 634: how are duplicate variants resolved in the variant union procedure? For example, variant VAF will likely differ between callers

Response: The merging procedure of variants called by multiple callers includes an in-house genotyping step that provides a unified calculation of variant depth of coverage, alt read support and VAF for each unique variant.

10. Line 652: More methodological description of the FACETS2n algorithm is required, as this is a major component of the paper. The R package is well documented, and installs, but is lacking a LICENSE file, technically rendering it currently unusable. Please provide example data and supporting files so that the package can be rapidly tested

Response: We thank the reviewer for these suggestions. A LICENSE file (GPL-3.0) has been added to the github repository. Regarding test data and commands for rapid testing, all of the commands and figures produced in the package Vignettes section, detailed in the README, can be reproduced with the existing data included with the package:

<https://github.com/mskcc/facets2n/tree/master/inst/extdata>

11. Line 737 The URL is dead, ensure this is active:

Response: We are working with our team to ensure the link is active as the publication goes online.

https://www.cbioportal.org/study/summary?id=heme_msk_impact_2022

12. Line 742 looks like a URL but is not.

Response: The hyperlink has been replaced with a full URL in the manuscript.

Reviewer #3 (Remarks to the Author):

The authors respond to my concerns about the limited additional value of paired tumor-normal sequencing (vs tumor-only), by explaining that private SNPs or putative passengers may be of potential clinical relevance. Whilst this remains possible and the examples the authors cite are valid (tracking the tumor, mutational signatures etc), these are not demonstrated in their study.

So whilst the manuscript describes a robust and affective platform, the technical or clinical advances demonstrated here are limited.

Response: We appreciate the comments raised by the reviewer. The key point that we hope to convey in our work is the immense utility and feasibility of implementation of routine paired tumor normal sequencing. To our knowledge, we are the only institution who has fully implemented this as part of routine clinical testing, prompting the implementation of both novel technical and workflow approaches to make this not only feasible but sustainable. We outline the key aspects of the clinical utility and highlight that as we go forward with whole genome and whole exome sequencing, the current model of inferring a germline variant from curated databases and VAF's would no longer be feasible. We also make our sequencing results available for public use and hope this will lay the ground-work for others to build upon. While the focus of this manuscript was as stated above, more detailed demonstrations of the application in patient subsets are actively being developed and we are excited to share them with the community when they are ready.

REVIEWERS' COMMENTS

Reviewer #1 (Remarks to the Author):

The authors have satisfactorily addressed my concerns.

Reviewer #2 (Remarks to the Author):

The authors have satisfactorily addressed my suggestions.

As <https://github.com/rptashkin/facets2n> doesn't contain a LICENSE file, I suggest you add a note to redirect investigators from <https://github.com/rptashkin/facets2n> to <https://github.com/mskcc/facets2n>